# Linear mode connectivity in multitask and continual learning

**Seyed Iman Mirzadeh**[*]
Washington State University, USA
`seyediman.mirzadeh@wsu.edu`

**Mehrdad Farajtabar**[*]
DeepMind, USA
`farajtabar@google.com`

**Dilan Gorur**
DeepMind, USA
`dilang@google.com`

**Razvan Pascanu**
DeepMind, UK
`razp@google.com`

**Hassan Ghasemzadeh**
Washington State University, USA
`hassan.ghasemzadeh@wsu.edu`

## ABSTRACT

Continual (sequential) training and multitask (simultaneous) training are often attempting to solve the same overall objective: to find a solution that performs well on all considered tasks. The main difference is in the training regimes, where continual learning can only have access to one task at a time, which for neural networks typically leads to catastrophic forgetting. That is, the solution found for a subsequent task does not perform well on the previous ones anymore. However, the relationship between the different minima that the two training regimes arrive at is not well understood. What sets them apart? Is there a local structure that could explain the difference in performance achieved by the two different schemes? Motivated by recent work showing that different minima of the same task are typically connected by very simple curves of low error, we investigate whether multitask and continual solutions are similarly connected. We empirically find that indeed such connectivity can be reliably achieved and, more interestingly, it can be done by a linear path, conditioned on having the same initialization for both. We thoroughly analyze this observation and discuss its significance for the continual learning process. Furthermore, we exploit this finding to propose an *effective algorithm* that constrains the sequentially learned minima to behave as the multitask solution. We show that our method outperforms several state of the art continual learning algorithms on various vision benchmarks.[1]

## 1 INTRODUCTION

One major consequence of learning multiple tasks in a continual learning (CL) setting — where tasks are learned sequentially, and the model can only have access to one task at a time — is catastrophic forgetting (McCloskey & Cohen, 1989). This is in contrast to multitask learning (MTL), where the learner has simultaneous access to all tasks, which generally learns to perform well on all tasks without suffering from catastrophic forgetting. This limitation hinders the ability of the model to learn continually and efficiently. Recently, several approaches have been proposed to tackle this problem. They have mostly tried to mitigate catastrophic forgetting by using different approximations of the multitask loss. For example, some regularization methods take a quadratic approximation of the loss of previous tasks (e.g. Kirkpatrick et al., 2017; Yin et al., 2020). As another example, rehearsal methods attempt to directly use compressed past data either by selecting a representative subset (e.g. Chaudhry et al., 2019; Titsias et al., 2019) or relying on generative models (e.g. Shin et al., 2017; Robins, 1995).

In this work, we depart from the literature and start from the non-conventional question of understanding *"What is the relationship, potentially in terms of local geometric properties, between the multitask and the continual learning minima?"*. Our work is inspired by recent work on mode con-

---

[*]Equal contribution
[1]The code is available at:`https://github.com/imirzadeh/MC-SGD`

nectivity (Draxler et al., 2018; Garipov et al., 2018; Frankle et al., 2020) finding that different optima obtained by gradient-based optimization methods are connected by simple paths of non-increasing loss. We try to understand whether the multitask and continual solutions are also connected by a manifold of low error, and what is the simplest form this manifold can take. Surprisingly, we find that a linear manifold, as illustrated in Fig. 1 right, reliably connects the multitask solution to the continual ones, granted that the multitask shares same initialization with the continual learning as described below. This is a significant finding in terms of understanding the phenomenon of catastrophic forgetting through the lens of loss landscapes and optimization trajectory and also for designing better continual learning algorithms.

To reach this conclusion, we consider a particular learning regime described in Fig. 1 left, where after learning the first task using the data $\mathcal{D}_1$, we either sequentially learn a second task obtaining $\hat{w}_2$ or continue by training on both tasks simultaneously (i.e., train on $\mathcal{D}_1 + \mathcal{D}_2$), obtaining the multitask solution $w_2^*$. We investigate the relationship between the two solutions $\hat{w}_2$ and $w_2^*$. Note that $w_2^*$ is not the typical multitask solution, which would normally start from $w_0$ and train on both datasets. We chose this slightly non-conventional setup to minimize the potential number of confounding factors that lead to discrepancies be-

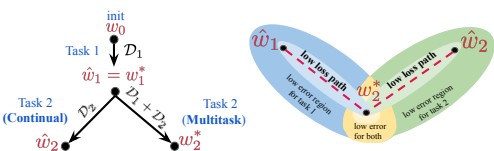

Figure 1: **Left:** Depiction of the training regime considered. First $\hat{w}_1$ is learned on task 1. Afterwards we either reach $\hat{w}_2$ by learning second task or $w_2^*$ by training on both tasks simultaneously. **Right:** Depiction of linear connectivity between $w_2^*$ and $\hat{w}_1$ and between $w_2^*$ and $\hat{w}_2$.

tween the two solutions (Fort et al., 2019). We also rely on the observation from (Frankle et al., 2020) that initialization can have a big impact on the connectivity between the solutions found on the same task, and sharing the same starting point, as we do between $\hat{w}_2$ and $w_2^*$, might warrant a linear path of low error between the two solutions. Moreover, Neyshabur et al. (2020) noted that in the context of transfer learning, there is no performance barrier between two minima that start from pre-trained weights, which suggests that the pre-trained weights guide the optimization to a flat basin of the loss landscape. In contrast, barriers clearly exist if these two minima start from randomly initialized weights.

Our contributions can be summarized as follows:

1. To the best of our knowledge, our work is the first to study the connectivity between continual learning and multitask learning solutions.

2. We show that compared to conventional similarity measures such as Euclidean distance or Central Kernel Alignment (Kornblith et al., 2019), which are incapable of meaningfully relating these minima, the connectivity through a manifold of low error can reliably be established. And this connecting path is linear, even when considering more than 20 tasks in a row.

3. Motivated by this, we propose an effective CL algorithm (Mode Connectivity SGD or MC-SGD) that is able to outperform several established methods on standard CL benchmarks.

## 1.1 RELATED WORK

With the trending popularity of deep learning, continual learning has gained a critical importance because the catastrophic forgetting problem imposes key challenges to deploy deep learning models in various applications (e.g Lange et al., 2019; Kemker et al., 2018). A growing body of research has attempted to tackle this problem in recent years (e.g Parisi et al., 2018; Toneva et al., 2018; Nguyen et al., 2019; Farajtabar et al., 2019; Hsu et al., 2018; Rusu et al., 2016; Li et al., 2019; Kirkpatrick et al., 2017; Zenke et al., 2017; Shin et al., 2017; Rolnick et al., 2018; Lopez-Paz & Ranzato, 2017; Chaudhry et al., 2018b; Riemer et al., 2018; Mirzadeh et al., 2020; Wallingford et al., 2020). Among these works, our proposed MC-SGD bares most similarities to *rehearsal* based methods such us (e.g. Shin et al., 2017; Chaudhry et al., 2018b) and *regularization* based methods (e.g. Kirkpatrick et al., 2017; Zenke et al., 2017) similar to (Titsias et al., 2019). Following (Lange et al., 2019), one can categorize continual learning methods into three general categories, based on how they approach dealing with catastrophic forgetting.

**Experience replay**: Experience replay methods build and store a memory of the knowledge learned so far (Rebuffi et al., 2016; Lopez-Paz & Ranzato, 2017; Shin et al., 2017; Riemer et al., 2018; Rios & Itti,

2018; Zhang et al., 2019). As examples, Averaged Gradient Episodic Memory (A-GEM) (Chaudhry et al., 2018b) builds an episodic memory of parameter gradients, while ER-Reservoir (Chaudhry et al., 2019) uses a Reservoir sampling method to maintain the episodic memory.

**Regularization**: These methods explicitly apply regularization techniques to ensure parameters do not change too much (Kirkpatrick et al., 2017; Zenke et al., 2017; Lee et al., 2017; Aljundi et al., 2018; Kolouri et al., 2019). They can either a Bayesian (Nguyen et al., 2017; Titsias et al., 2019; Schwarz et al., 2018; Ebrahimi et al., 2019; Ritter et al., 2018) or frequentist views (Farajtabar et al., 2019; He & Jaeger, 2018; Zeng et al., 2018). For instance, Orthogonal Gradient Descent (OGD) (Farajtabar et al., 2019) projects the prediction gradients from new tasks on the subspace of previous tasks' gradients to preserve the knowledge.

**Parameter isolation**: Parameter isolation methods allocate different subsets of the parameters to each task (Rusu et al., 2016; Yoon et al., 2018; Jerfel et al., 2019; Rao et al., 2019; Li et al., 2019). From the stability-plasticity perspective, these methods implement gating mechanisms that improves the stability and controls the plasticity by activating different gates for each task. Masse et al. (2018) proposes a bio-inspired approach for a context-dependent gating that activates non-overlapping subset of parameters for any specific task. Supermask in Superposition (Wortsman et al., 2020) is another parameter isolation method that starts with a randomly initialized, fixed base network and for each task finds a subnetwork (supermask) such that the model achieves good performance. Recently, Mirzadeh et al. (2020) have shown that dropout implicitly creates different pathways or gates for tasks, therefore, it reduces their mutual interference and leads to less forgetting.

Continual learning as a problem expands beyond dealing with catastrophic forgetting, one of the hopes behind sequential learning is that it can potentially enable positive forward transfer, as one can build on the previously acquired knowledge. In this sense it connects to topics such as Meta-Learning (Beaulieu et al., 2020; Jerfel et al., 2019; He & Jaeger, 2018; Riemer et al., 2018), Few-Shot Learning (Wen et al., 2018; Gidaris & Komodakis, 2018), Multi-task and Transfer Learning (He & Jaeger, 2018; Jerfel et al., 2019). It also aims to work in scenarios where task boundaries are not well defined or provided, or when the data distribution shifts slowly or when a multi-task solution does not exist (Rao et al., 2019; Aljundi et al., 2019; He et al., 2019; Kaplanis et al., 2019).

**Mode connectivity** (Draxler et al., 2018; Garipov et al., 2018), is a novel tool to understand the loss landscape of deep neural networks. It postulates that different optima obtained by gradient-based optimization methods are connected by simple paths of non-increasing loss (i.e., low-loss valleys). Recently, various works provided different theoretical explanations for mode connectivity in different scenarios (Venturi et al., 2019; Kuditipudi et al., 2019) either by relying on over-parametrization or concepts such as noise-stability or dropout-stability. (Neyshabur et al., 2020) investigated the connection between minima obtained by pre-trained models versus freshly initialized ones. They note that there is no performance barrier between solutions coming from pre-trained models, but there can be a barrier between solutions of different randomly initialized models. (Frankle et al., 2020) shows that different minima that share the same initialization point are connected by a linear path, even with weight pruning. We rely on this observation when designing our setting that the multitask and continual learning minima share a common starting point.

## 2 THE RELATION BETWEEN MULTITASK AND CONTINUAL MINIMA

One question driving this work is understanding the relationship between the two different solutions: multitask learning vs. continual learning. In particular, we are interested in scenarios where a multitask solution for the considered tasks exists (for a discussion when this does not hold see (He et al., 2019)) and when both learning regimes have the same objective, finding a solution that performs well on all tasks. We posit that this difference can not be reliably explained by simple and typical distance metrics used to measure similarity between the trained models. In particular we consider Euclidean distance and Central Kernel Alignment (CKA) (Kornblith et al., 2019). However, these solutions are connected by paths of low error, and, provided that learning starts from the same initial conditions, these paths can have a linear form.

In Fig. 2 left column we can see the performance of Naive SGD for multitask and continual learning. Details on the experimental setup can be found in the Appendix C. The dashed line represents the multitask solution at convergence, which achieves strong performance on all tasks. It further shows the performance of all tasks during the sequential learning experience (each point represents the

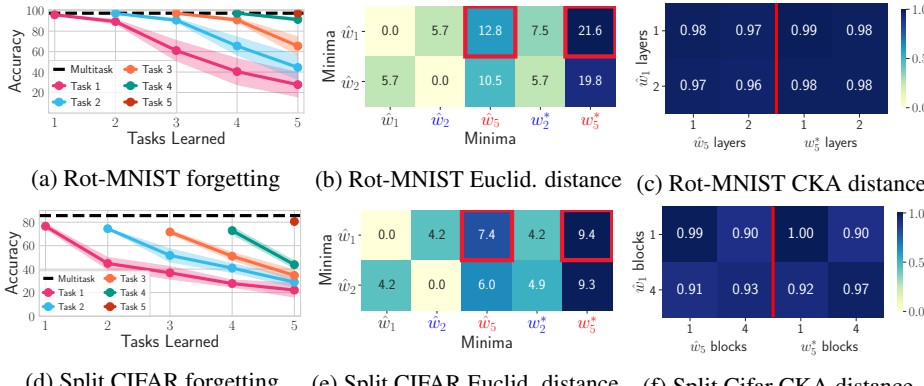

(a) Rot-MNIST forgetting   (b) Rot-MNIST Euclid. distance   (c) Rot-MNIST CKA distance

(d) Split CIFAR forgetting   (e) Split CIFAR Euclid. distance   (f) Split Cifar CKA distance

Figure 2: Continual and Multitask learning performance and relation between minima. Top row: Rotation MNIST. Bottom row: Split CIFAR-100. **Left column**: Validation accuracy of all tasks during continual training. **Middle**: Euclidean distance: note that $\hat{w}_5$ is not a good solution for task 1 even though it is closer (more similar) to $\hat{w}_1$ than $w_5^*$ in Euclidean distance. **Right**: CKA distance: $\hat{w}_5$ is closer (more similar) to $\hat{w}_1$ than $w_5^*$ in terms of CKA distance. Therefore, neither Euclidean nor CKA distance is able to realize MTL is better in avoiding catastrophic forgetting.

performance after learning another task), highlighting how performance on past tasks degrades considerably. Note that as described in Fig. 1 and further detailed in Appendix D.2 Fig. 15, in the multitask learning scenario, tasks are added sequentially to the loss, to construct a parallel with the continual learning setting. This will be the case throughout this work.

**Eucledian distance**. It might be reasonable to expect that the less the parameter changes, the less the forgetting will be. One can motivate this heuristic on a Taylor expansion of the loss, as done in (Mirzadeh et al., 2020), where, the forgetting is defined as:

$$L_1(\hat{w}_2) - L_1(\hat{w}_1) \approx \frac{1}{2}(\hat{w}_2 - \hat{w}_1)^\top \nabla^2 L_1(\hat{w}_1)(\hat{w}_2 - \hat{w}_1) \leq \frac{1}{2}\lambda_1^{\max}\|\hat{w}_2 - \hat{w}_1\|^2. \quad (1)$$

Here, $L_1(w)$ is the empirical loss for task 1, and $\lambda_1^{\max}$ is the largest eigenvalue of its Hessian at $\hat{w}_1$. Note that all terms of the Taylor expansion are multiplied with $\hat{w}_2 - \hat{w}_1$ and hence the norm of this delta will affect the amount of forgetfulness. In fact, this is frequently done when pruning neural networks (Zhu & Gupta, 2018; Han et al., 2015), where weights are zeroed out based on magnitude, producing minimal Euclidean distance to unprunned model. But, as observed in Fig. 2 (middle column), Eucledian distance does not correlate with not suffering from catastrophic forgetting: the CL solution on task 1 ($\hat{w}_1$) is closer to the CL solution of task 5 ($\hat{w}_5$) than the multitask solution of tasks 5 ($w_5^*$). One explanation could be that the bound defined by Eq. (1) will not be tight if the vector $\hat{w}_5 - \hat{w}_1$ does not lie in the direction of the largest eigenvector. Appendix D.1 contains further details on this topic.

**Centered Kernel Alignment**. Centered Kernel Alignment (CKA) (Kornblith et al., 2019) measures the similarity of two representations on the same set of examples. Given $N$ examples and two activation outputs on these examples, $R_1 \in \mathbb{R}^{N \times d_1}$ and $R_2 \in \mathbb{R}^{N \times d_2}$, CKA is defined by:

$$\text{CKA}(R_1, R_2) = \frac{\|R_1^\top R_2\|_F^2}{\|R_1^\top R_1\|_F \|R_2^\top R_2\|_F}, \quad (2)$$

where, $\|.\|_F$ is the Frobenius norm. Recent work by Ramasesh et al. (2020) studies catastrophic forgetting on the CIFAR dataset by measuring the CKA similarity score of different layers of $\hat{w}_1$ and $\hat{w}_2$. They argue that the later layers suffer more from catastrophic forgetting by showing that the CKA similarity of initial layers decreases less after training on sequential tasks.

However, the CKA score suffers from a few shortcomings. If the number of training epochs per task is small (e.g., in streaming case), the CKA does not change much, even though the accuracy for previous tasks drops drastically. For instance, in Fig. 2 right column, we show that the pairwise CKA between different layers of the first task minimum ($\hat{w}_1$) and CL and multitask minima of task 2 and task 5 are roughly the same. Although, the phenomenon observed in (Ramasesh et al., 2020) is still realizable by a very tiny margin. Moreover, we can see that the multitask minimum of task 2 ($w_2^*$), and task 5 ($w_5^*$) are more similar to $\hat{w}_1$ compared to CL minima ($\hat{w}_2$ and $\hat{w}_5$).

## 2.1 MODE CONNECTIVITY

Mode connectivity has been studied empirically (Draxler et al., 2018; Garipov et al., 2018) and theoretically with some assumptions (e.g., over-parameterization or noise stability) (Kuditipudi et al., 2019; Venturi et al., 2019). We note that these previous works were focused on "single-task" setup, where the data stays $i.i.d.$, and more importantly, the networks start from different initialization. Because of this assumption, it is shown that the minima are not linearly connected (*i.e.*, the loss is not necessarily decreasing in the interpolation line between the two minima). Instead, the path consists of several line-segments or curves (Kuditipudi et al., 2019).

However, between our continual and multitask settings, a more favorable property can exist, namely that learning starts from the same initialization (i.e., $\hat{w}_1 = w_1^*$). For a "single task", common initialization leads to linear path between minima (Frankle et al., 2020). Therefore, our first step is to investigate whether the same linear connectivity holds between multitask and CL solutions. As we show in Fig. 3, this is true for both MNIST and CIFAR benchmarks. In this figure, we can see that the multitask minima for task 2 ($w_2^*$) is connected to both task 1 CL minima ($\hat{w}_1$) and task 2 CL minima ($\hat{w}_2$). We note that the surface plots in Fig. 3 are not of low-dimensional projection types and are rather "interpolation plots" in parameter space computed on the hyper-plane connecting these three minima. More details are provided in Appendix C.5.

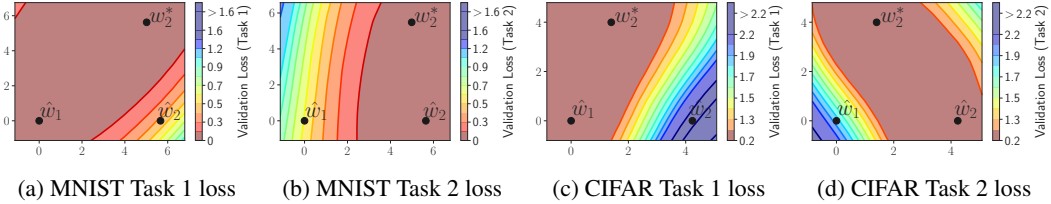

(a) MNIST Task 1 loss    (b) MNIST Task 2 loss    (c) CIFAR Task 1 loss    (d) CIFAR Task 2 loss

Figure 3: Cross-entropy validation loss surface of on rotation MNIST (a and b), and split CIFAR-100 (c and d), as a function of weights in a two-dimensional subspace passing through $\hat{w}_1$, $\hat{w}_2$, and $w_2^*$.

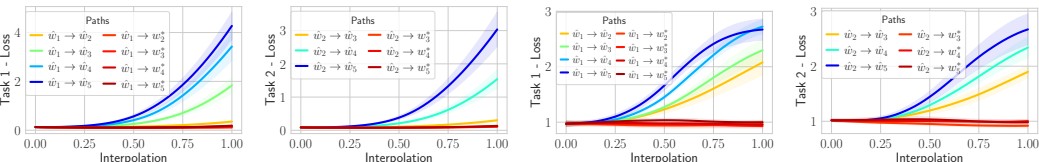

(a) MNIST $\hat{w}_1$ to next ones (b) MNIST $\hat{w}_2$ to next ones (c) CIFAR $\hat{w}_1$ to next ones (d) CIFAR $\hat{w}_2$ to next ones

Figure 4: Exploring the loss along the linear paths connecting the different solutions: The loss increases on the interpolation line between the first task solution($\hat{w}_1$) and subsequent continual solutions, while the loss remains low on the interpolation line between $\hat{w}_1$ and subsequent multitask minima (a and c). The same observation also holds for the second task solution ($\hat{w}_2$) (b and d)

To demonstrate linear connectivity holds for subsequent MTL and CL minima, we plot the validation loss on the line-segment interpolating between all subsequent minima for both task 1 and 2 on Rotated MNIST and CIFAR in Fig. 4. The low loss interpolation to $w^*$'s indicates that indeed along this linear path, the loss stays low.

## 3 WHEN AND WHERE DOES LINEAR MODE CONNECTIVITY HOLD?

The fact that linear regions of low error connect MTL and CL minima does not necessarily imply it is trivial to find one such path. In this section, we try to analyze this finding through the geometry of loss functions and, more specifically, its curvature, which identifies the amount of change when we move from solution of task 1 ($\hat{w}_1$) to either MTL ($w_2^*$) or CL ($\hat{w}_2$) solution. This way, we can explain why existing methods such as EWC that are also trying to penalize the change in directions of high curvature fail to find a plausible direction. The short answer is that they only rely on the second-order Taylor approximation of the loss function, which may be easily violated. We elaborate as follows.

Let us start by trying to understand what is implied by the fact that these linear paths exist. Focusing on the linear path between $\hat{w}_1$ and $w_2^*$, one potential justification is that while minimizing the multitask

loss, the updates move us in the direction of low curvature for the Hessian of the first loss around $\hat{w}_1$. To see why, let us make the mild assumption that the function is smooth, which typically tends to hold in the over-parameterized regime (e.g., similar assumptions are needed for the NTK regime to hold (Bietti & Mairal, 2019)). Then we can not have a rapid change in the curvature. Hence, if moving along the line connecting $\hat{w}_1$ and $w_2^*$, the loss of task 1 stays very small, it means the curvature also has to be small as it can not show rapid fluctuations that will not be reflected in the loss. Implicitly the curvature is low at $\hat{w}_1$ as well; hence this direction necessarily has to be within the subspace spanned by the eigenvectors with low curvature. This seems not to hold for $\hat{w}_1$ and $\hat{w}_2$, so we might expect that the line connecting them to move in directions of higher curvature. A symmetric argument can be made for $w_2^*$ and $\hat{w}_2$, looking now at the Hessian and loss of task 2. The smoothness argument makes it unlikely that curvature shows rapid fluctuations; hence loss staying low implies the alignment with eigenvectors of low curvature.

Before moving forward, let us empirically validate the argument in the previous paragraph. To do so, we estimate the top 50 eigenvalues/eigenvectors of the Hessian matrix around task 1 minima of rotation MNIST benchmark, similar to the setting we discussed in Section 2. We use the deflated power iteration method to obtain the eigenspectrum of the Hessian. Fig. 5a shows the largest eigenvalues. We note that similar to the observation by Fort & Ganguli (2019) and Mirzadeh et al. (2020), the eigenspectrum of the Hessian includes a bulk plus $C$ larger outlier eigenvalues where $C$ is the number of classes. To quantify the confinement between the subspace spanned by the Hessian eigenvectors and the movement directions, we measure the cosine angle of these directions and the Hessian eigenvectors. Fig. 5b shows that the direction from $\hat{w}_1$ to $\hat{w}_2$ lies within the subspace spanned by the top eigenvectors. Note that this agrees with our hypothesis. However, the direction between $\hat{w}_1$ to $w_2^*$ does not lie within this subspace as the cosine angle between this direction and the eigenvectors are all near zero, suggesting that this direction is orthogonal to all these eigenvalues and hence is spanned by eigenvectors with low eigenvalues. The orthogonality has also been tried as a hard constraint in a number of continual learning papers explicitly (Farajtabar et al., 2019; He & Jaeger, 2017). This explains the less suffering from catastrophic forgetting as we move in directions of low curvature.

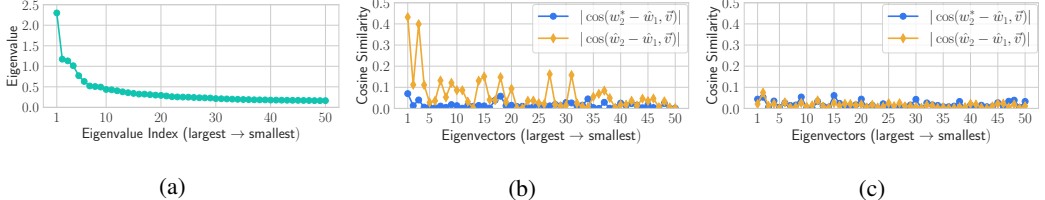

(a)            (b)            (c)

Figure 5: Comparison of the eigenspectrum of the Hessian matrix for $\hat{w}_1$. (a): top Eigenvalues. (b and c): The overlap between Hessian eigenvectors and directions from $\hat{w}_1$ to $\hat{w}_2$ and $w_2^*$ within the suitable region that 2nd order Taylor approximation holds (b) and does not hold (c).

Now, let us examine whether moving in the direction of low curvature is sufficient in addition to being a necessary condition or not? Fig. 5c shows that this is not sufficient. By increasing the number of epochs from 5 to 20 we show that both directions to $\hat{w}_2$ and $w_2^*$ do not lie in this subspace of the top eigenvectors. That is, for both, if we move on the segment away from $\hat{w}_1$, we are aligned with none of the directions associated with high curvature. But $\hat{w}_2$ suffers from catastrophic forgetting and hence a considerable increase in the loss of task 1, while $w_2^*$ does not. This suggests that while initially moving towards $\hat{w}_2$ the curvature stays low, but eventually higher-order terms increase the curvature of task 1 loss before reaching $\hat{w}_2$. Overall this implies the multiple $w^*$ that we observe, which gradually solve more and more tasks, lie within a region where a second-order Taylor approximation of the first task still holds, and higher-order derivatives do not play an important role. And within this region, learning is restricted to the subspace spanned only by eigenvectors corresponding to small eigenvalues for all previous tasks.

We empirically show that we can learn *up to 20 complex tasks and up to 50 permuted MNIST tasks in this regime* (see Fig. 7 here and 20 in Appendix D.5), hence it seems this region, where the second-order Taylor approximation holds, is quite expressive and sufficient to learn many tasks without interference between them. Additionally, similar linearizations seem to hold reliably in over parametrized models throughout learning as shown by the NTK-based literature (e.g Jacot et al., 2018; Bietti & Mairal, 2019) and these assumptions are implicitly made by several successful continual

learning algorithms (e.g. Kirkpatrick et al., 2017; Farajtabar et al., 2019; Mirzadeh et al., 2020). In particular, this observation is interesting when viewed from the perspective of regularization-based CL methods. Many of these methods rely on a second-order Taylor expansion (if not of previous losses then of the KL term with respect to how much the model changed) and forbid change in weights with high curvature. The fact that multitask learning adheres to the same principles, moves in the direction of low curvature and restricts itself to the region where this approximation holds, suggests that this remains a promising direction. It also highlights the importance of better approximations of curvature and the importance of remaining within the area where this approximation holds.

In Appendix B, we further investigate different aspects of our setting that might make it possible to find reliable solutions for all subsequent tasks in an area around the solution of the first task where the second-order Taylor approximation holds. We show, that as assumed earlier, sharing the same initial condition is vital for linear connectivity to occur. Furthermore, the shared structure between tasks like low-level features in images or semantically consistent labeling plays an important role as well. If we destroy any structure between tasks, we can not find a multitask solution for task 1 and 2 that is linearly connected to the first task.

## 4 CONTINUAL LEARNING WITH MODE CONNECTIVITY

Based on the observation in Section 2.1 that there exist minima with linear mode connectivity to previous CL minima, in this section, we propose a continual learning algorithm that exploits this property. The proposed algorithm can be viewed as a combination of both replay and regularization methods in continual learning.

### 4.1 MC-SGD

Mode Connectivity SGD (MC-SGD) is a light-weight method that constrains the continual minima to lie in a low-loss valley to all previous minima. For simplicity, we start by providing the loss function for two tasks, then we extend it for an arbitrary number of tasks. We define the loss objective that enforces linear connectivity to both $\hat{w}_1$ and $\hat{w}_2$ as below:

$$\bar{w} = \arg\min_w \int_{0 \leq \alpha \leq 1} [\mathcal{L}_1(\hat{w}_1 + \alpha(w - \hat{w}_1)) + \mathcal{L}_2(\hat{w}_2 + \alpha(w - \hat{w}_2))] \, d\alpha, \qquad (3)$$

where $\mathcal{L}_1$ and $\mathcal{L}_1$ are the loss function for task 1 and task 2 respectively and $\alpha$ parameterizes the line connecting $w$ to $\hat{w}_1$ and $\hat{w}_2$ respectively in the first and second term. Essentially, Eq. (3) constrains the MC-SGD minima to have a low-loss path to both $\hat{w}_1$ and $\hat{w}_2$. The integral in this equation can be approximated by averaging the loss over a few random $\alpha$ in $[0, 1]$. However, our experiments showed that picking as few as $n = 5$ equally spaced points between 0 and 1 is more than sufficient to get good results. We can further decompose (3) into:

$$\bar{w} = \arg\min_w \quad \mathcal{L}_1(w) + \mathcal{L}_2(w) + \underbrace{\mathcal{L}_1(\hat{w}_1) + \mathcal{L}_2(\hat{w}_2)}_{\text{constant}}$$

$$+ \underbrace{\frac{1}{n-1} \sum_{\alpha \in \{\frac{1}{n}, \ldots, \frac{n-1}{n}\}} [\mathcal{L}_1(\hat{w}_1 + \alpha(w - \hat{w}_1)) + \mathcal{L}_2(\hat{w}_2 + \alpha(w - \hat{w}_2))]}_{\text{regularization}} \quad (4)$$

It's evident from Eq. (4) that $\bar{w}$ should minimize both task 1 and task 2 loss, in addition to having low-loss paths to both of the CL minima.

In a continual learning setting, however, we would not have access to $\mathcal{L}_1$ once we start learning on $\mathcal{L}_2$. Therefore we rely on experience replay like ER (Chaudhry et al., 2019) and AGEM (Chaudhry et al., 2018b). We use a small replay buffer with randomly sampled examples to approximate $\mathcal{L}_1$, with the hope that given the restricted form of Eq. (4) through the additional regularizer, we might need lesser data then if we would have used it to approximate directly the multitask loss.

We can further extend the mode connectivity loss to an online version that is cheap to compute:

$$\bar{w}_t = \arg\min_w \quad \sum_\alpha [\mathcal{L}_{t\text{-}1}(\bar{w}_{t-1} + \alpha(w - \bar{w}_{t-1})) + \mathcal{L}_t(\hat{w}_t + \alpha(w - \hat{w}_t))]. \qquad (5)$$

Eq. (5) can be viewed as follows: when learning task $t$, assuming that we have a high-performance solution for the previous $t - 1$ tasks (i.e., $\bar{w}_{t-1}$), we try to minimize Eq. (3) as if in this equation, $\hat{w}_1$ is $\bar{w}_{t-1}$ and $\hat{w}_2$ is $\hat{w}_t$. The replay memory still consists of samples from all the previous $t - 1$ tasks.

## 5    EXPERIMENTS AND RESULTS

The experimental setup, such as benchmarks, network architectures, continual learning setting (e.g., number of tasks, episodic memory size, and training epochs per task), hyper-parameters, and evaluation metrics are chosen to be similar to several other studies (Chaudhry et al., 2018b; Mirzadeh et al., 2020; Chaudhry et al., 2019; Farajtabar et al., 2019; Chaudhry et al., 2019). For all experiments, we report the average and standard deviation over five runs with different random seeds. Appendix C provides a detailed discussion on our experimental setup explained below.

**Benchmarks.** We report on three standard continual learning benchmarks: Permuted MNIST (Goodfellow et al., 2013), Rotated MNIST, and Split CIFAR-100. Although we are aware of the shortcomings of Permuted MNIST (Farquhar & Gal, 2018), for the sake of consistency with literature, we report the result on this benchmark. However, we also report our results on Rotated MNIST and CIFAR-100, that are more challenging and realistic datasets for continual learning, especially when the number of tasks is large. Each task of permuted MNIST is generated by random shuffling of the pixels of images such that the permutation remains the same for the images of within the same task, but changes across different tasks. Rotated MNIST is generated by the continual rotation of the MNIST images where each task applies a fixed random image rotation (between 0 and $180°$). Split CIFAR is a variant of CIFAR-100 where each task contains the data from 5 random classes.

**Architectures.** For the rotation and permutation MNIST benchmarks, we use a two-layer MLP with 256 ReLU units in each layer. For the split CIFAR-100 benchmark, we use a ResNet18, with three times fewer feature maps across all layers.

**Evaluation.** Following several studies in the literature (Chaudhry et al., 2018b; Mirzadeh et al., 2020), we report two metrics to compare CL algorithms when the number of tasks is large.
(1) Average Accuracy: The average validation accuracy after the model has been continually learned task $t$, is defined by $A_t = \frac{1}{t} \sum_{i=1}^{t} a_{t,i}$, where, $a_{t,i}$ is the validation accuracy on dataset $i$ after the model finished learning task $t$.
(2) Average Forgetting: The average forgetting measures the backward transfer in continual learning. This metric is calculated by the difference of the peak accuracy and ending accuracy of each task, after the continual learning experience has finished. For a continual learning benchmark with $T$ tasks, it is defined by $F = \frac{1}{T-1} \sum_{i=1}^{T-1} \max_{t \in \{1,...,T-1\}} (a_{t,i} - a_{T,i})$.

### 5.1    COMPARISON WITH OTHER METHODS

In this experiment, we compare MC-SGD with various established continual learning algorithms. Our setup consists of 20 tasks for three benchmarks: permuted MNIST, Rotated MNIST, and split CIFAR-100. The episodic memory size for A-GEM, ER-Reservoir, and MC-SGD is limited to be one example per class per task. Table 1 compares the average accuracy and average forgetting (i.e., $A_t$ and $F$ defined above) for each method once the continual learning experience is finished (i.e., after learning task 20). Moreover, Fig. 6 shows the evolution of average accuracy for each method on each benchmark. We can see in the figure that MC-SGD outperforms other methods with its performance gap increasing as the number of tasks increases. In Appendix D.5, we show this trend also holds when the number of tasks increases from 20 to 50 for Permuted MNIST benchmark. Overall, the results validate a few aspects of our method. MC-SGD seems to make more efficient use of the episodic memory compared to replay based techniques A-GEM and ER-Reservoir, perhaps due to incorporating the additional knowledge of linear connectivity as a regularizer. Moreover, if we see our approach as restricting learning in the subspace of low curvature, this subspace is better approximated compared to EWC, which takes an explicit Taylor approximation of the KL term and adds a similar constraint (that learning can not change weights with high curvature).

Table 1: Comparison between the proposed method (MC-SGD) and other baselines.

| Method | Permuted MNIST | | Rotated MNIST | | Split CIFAR-100 | |
|---|---|---|---|---|---|---|
| | Accuracy ↑ | Forgetting ↓ | Accuracy ↑ | Forgetting ↓ | Accuracy ↑ | Forgetting ↓ |
| Naive SGD | 44.4 (±2.46) | 0.53 (±0.03) | 46.3 (±1.37) | 0.52 (±0.01) | 40.4 (±2.83) | 0.31 (±0.02) |
| EWC (Kirkpatrick et al., 2017) | 70.7 (±1.74) | 0.23 (±0.01) | 48.5 (±1.24) | 0.48 (±0.01) | 42.7 (±1.89) | 0.28 (±0.03) |
| A-GEM (Chaudhry et al., 2018b) | 65.7 (±0.51) | 0.29 (±0.01) | 55.3 (±1.47) | 0.42 (±0.01) | 50.7 (±2.32) | 0.19 (±0.04) |
| ER-Reservoir (Chaudhry et al., 2019) | 72.4 (±0.42) | 0.16 (±0.01) | 69.2 (±1.10) | 0.21 (±0.01) | 46.9 (±0.76) | 0.21 (±0.03) |
| Stable SGD (Mirzadeh et al., 2020) | 80.1 (±0.51) | 0.09 (±0.01) | 70.8 (±0.78) | 0.10 (±0.02) | 59.9 (±1.81) | 0.08 (± 0.03) |
| MC-SGD (ours) | **85.3** (±**0.61**) | **0.06** (±**0.01**) | **82.3** (±**0.68**) | **0.08** (±**0.01**) | **63.3** ( ±**2.21**) | **0.06** (± **0.03**) |
| Multitask Learning | 89.5 (±0.21) | 0.0 | 89.8(±0.37) | 0.0 | 68.8(±0.72) | 0.0 |

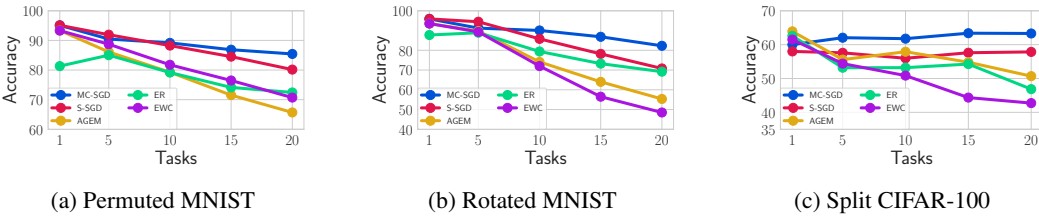

(a) Permuted MNIST       (b) Rotated MNIST       (c) Split CIFAR-100

Figure 6: Evolution of average accuracy during the continual learning experience with 20 tasks.

## 5.2 MODE CONNECTIVITY OF THE MC-SGD MINIMA

Next, we show that by minimizing Eq. (5), the minima found by MC-SGD are almost linearly connected to both the starting point of learning a new task (which represents the multitask solution of all previous task) as well as the solution of the task being learned. Fig. 7 shows the interpolation plots for rotated MNIST and split CIFAR-100. In Fig. 7a, we show the validation loss for task 1 on the interpolation line between $\hat{w}_1$ and four subsequent MC-SGD, and continual minima obtained during the learning of all subsequent tasks. We choose $\hat{w}_5$ and $\bar{w}_5$ as minima in the early stages of learning, $\hat{w}_{10}$ and $\bar{w}_{10}$ for the middle stages of learning, and $\hat{w}_{15}$, $\bar{w}_{15}$, $\hat{w}_{20}$, and $\bar{w}_{20}$ for the late stages of learning in our illustrations. We can see the losses on the interpolation lines between CL and MC minima are nearly flat compared to the losses on the lines among CL minima. Moreover, Fig. 7b shows the interpolation plot for task 5 to make sure the conclusion holds for later minima as well. Similarly, Figs. 7c and 7d show the interpolation plots of split CIFAR-100 for task 1 and 5.

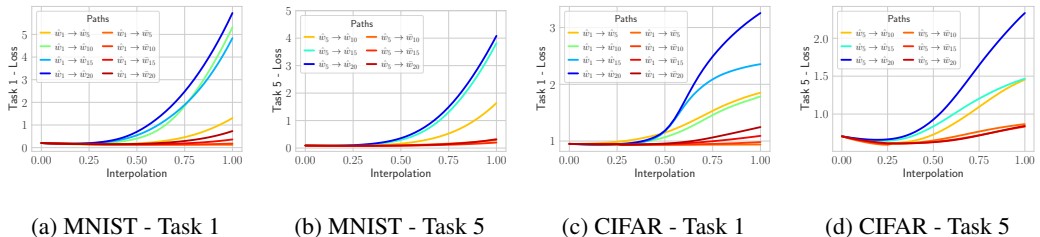

(a) MNIST - Task 1     (b) MNIST - Task 5     (c) CIFAR - Task 1     (d) CIFAR - Task 5

Figure 7: Mode Connectivity between the CL minima found by Naive SGD and minima found by the proposed MC-SGD: (a) and (b) Rotated MNIST , (c) and (d) Split CIFAR-100. In each figure, the left legend column corresponds to paths between Naive SGD minima, and the right column corresponds to paths between MC-SGD minima.

## 6 CONCLUSION

While both continual and multitask learning aim to find solutions that perform well across all tasks, the difference in their training regimes leads to two very different outcomes: multitask learning typically ends up achieving this goal, while continual learning suffers from catastrophic forgetting, performing well only on the recently seen tasks. In this work we investigated the relationship between these two solutions when all other potential confounding factors are minimized.

We considered a simple training regime in which multitask learning and continual learning start from similar conditions when incorporating a new task. We showed that in this condition, multitask minima are connected to continual learning ones by a linear path of low error, while continual learning solutions are not similarly connected. This can be understood through the glance of Taylor expansions as multitask objective restricts learning in the directions of low curvature within an area where a second order approximation holds. Such solutions seem to exists even when the process is applied repeatedly, solving more than 20 tasks in a row.

Finally, we explored this observation and proposed a new algorithm for continual learning called Mode Connectivity SGD (MC-SGD). It relies on the assumption that there always exist a solution able to solve all seen tasks so far that is connected by a linear path of low loss. MC-SGD utilizes a replay buffer to approximate the loss of for previous tasks. However, compared to other popular rehearsal-based methods, it performs better with less data as it exploits the linear path to further constrain learning.

ACKNOWLEDGEMENTS

SIM and HG acknowledge support from the United States National Science Foundation through grant CNS-1750679. The authors thank Jonathan Schwarz, and Behnam Neyshabur for their valuable comments and feedback.

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

# A  OUTLINE

The appendix is organized as follows:

**Appendix B** provides detailed discussion on the required assumptions for mode connectivity.

**Appendix C** gives a comprehensive discussion on the experimental setup for each experiment, hyper-parameters, visualization, and implementation. In addition, we provide anonymous links to the code and details to reproduce the results.

**Appendix D** contains additional experiments and results, including detailed plots for mode connectivity of continual and multitask minima in Sec. 2, and an additional experiment on permuted MNIST dataset with 50 tasks.

# B  BREAKING THE LINEAR CONNECTIVITY BETWEEN THE CONTINUAL LEARNING AND MULTITASK SOLUTION

## B.1  DIFFERENT INITIALIZATION

Furthermore, it is useful to understand if we can break the linear path between the multitask and continual learning minima. An initial assumption was that this is caused by the sharing of the same initialization, where both $\hat{w}_2$ and $w_2^*$ are learned starting from $\hat{w}_1$. Fig. 8 shows how violating this assumption breaks the linear mode connectivity for the split CIFAR-100 benchmark. Fig. 8a shows the validation loss interpolation between $\hat{w}_1$ and the the next four multitask and CL minima. As illustrated, the multitask minima is no longer linearly connected to $\hat{w}_1$ and is separated by a wall. The same phenomenon holds for the next minima, and in Fig. 8c, we show this for $\hat{w}_2$. We note that for MNIST datasets, as reported by Frankle et al. (2020), the linear connectivity still holds even if the MTL minimum starts from other initialization points. Related to this, Neyshabur et al. (2020) investigated the connection between minima obtained by pre-trained models versus freshly initialized ones. They pointed out that there is no performance barrier between solutions coming from pre-trained models, but there can be a barrier between solutions of different randomly initialized models. Here, $\hat{w}_1$ acts like a pre-trained model for multitask objective, placing the optimizer in a good basin for task 1.

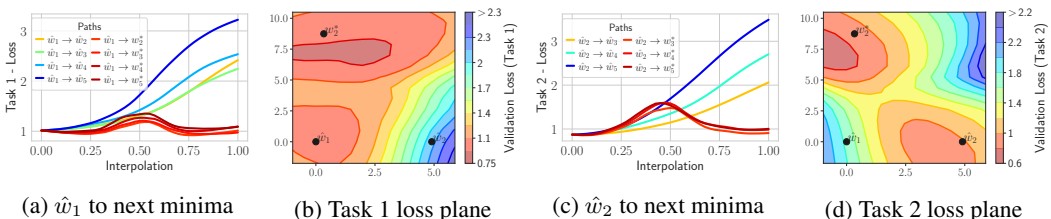

(a) $\hat{w}_1$ to next minima     (b) Task 1 loss plane     (c) $\hat{w}_2$ to next minima     (d) Task 2 loss plane

Figure 8: The linear connectivity does not hold for MTL minima that start from different initializations on CIFAR-100

## B.2  SHARED STRUCTURE OR SEMANTICS OF DATA

Another potentially important role might be played the shared structure between tasks. If the different tasks do not share a common structure it might be unlikely that there exists a solution for one task in the neighborhood of the other. To manipulate the shared structure of tasks, we can either corrupt input images or corrupt the labels. To increase the distribution shift across tasks, in this experiment, we use a continual learning setup with two tasks: The first task is the MNIST dataset, and the second task is the Fashion-MNIST dataset.

To corrupt the input data, we add different amounts of independent Gaussian noise to the input images of the second task (i.e, Fashion MNIST) and report the results in Fig. 9. Fig. 9b to 9d show the gradual separation of $w_2^*$ and $\hat{w}_2$ as the amount of noise increases. The validation loss values on the linear interpolation between these two minima are also shown in Fig. 9a.

Fig. 10b to 10d show the effect of randomly changing the labels for different portions of the images of task 2. As illustrated, the more labels we corrupt, the more disconnected $\hat{w}_2$ and $w_2^*$ become. The loss values on the linear interpolation path between these minima are shown in Fig. 10a for different percentage of label corruption.

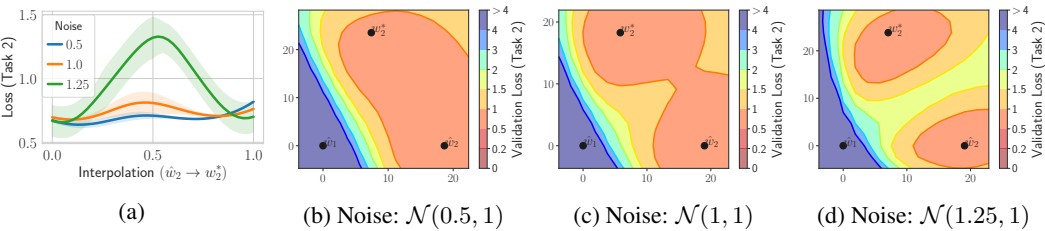

Figure 9: Noise Injection: **(a)**: The interpolation on task 2 for different amount of Gaussian noise on images, **(b)**, **(c)**, and **(d)**: $w_2^*$ becomes disconnected from $\hat{w}_2$ as the amount of noise increases

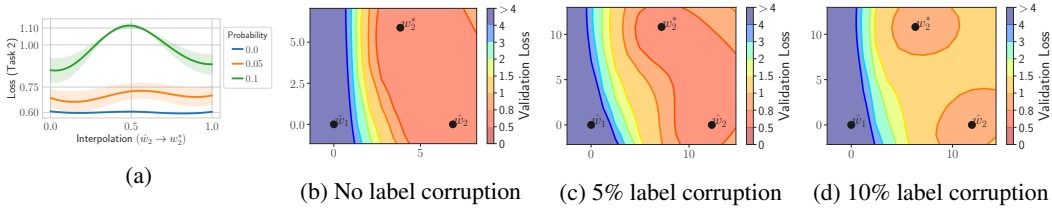

Figure 10: Label Corruption: **(a)**: The interpolation on task 2 for different amount of label corruption, **(b)**, **(c)**, and **(d)**: $w_2^*$ becomes disconnected from $\hat{w}_2$ as the more labels are corrupted

Yet a very trivial case of breaking mode connectivity is to cut a few classes from the second task. To this end, we perform the following experiment: After training on the first task and obtaining $\hat{w}_1$, for the multitask training on the second task, we randomly choose $k$ classes out of the 10 classes and remove the first task's examples that belong to any of these $k$ classes from the training dataset of multitask model.

In Fig. 11a, we show that by removing different number of classes from the training set of multitask model, the linear mode connectivity does not hold anymore as the validation loss on task 1 increases on the line between $\hat{w}_1$ and $w_2^*$. Moreover, while without removing any classes we have linear mode connectivity as illustrated in Fig. 11b, by removing 2 classes, $\hat{w}_2$ is no longer connected to $\hat{w}_1$ (Fig. 11c). The trend continues if we remove more classes (i.e, 4 classes in Fig. 11d).

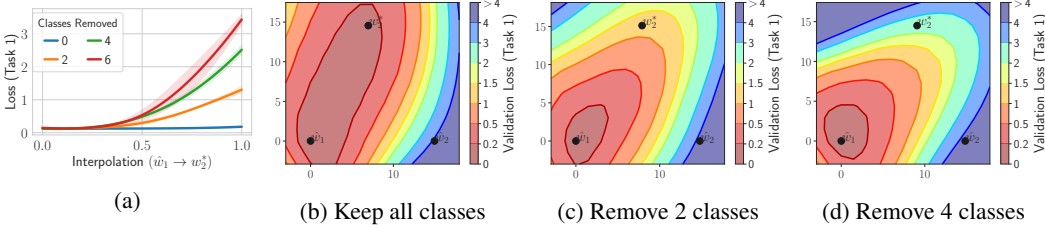

Figure 11: Removing Classes: **(a)**: Interpolation , **(b)**, **(c)**, and **(d)**: $w_2^*$ becomes more disconnected as we remove more and more classes

We emphasize that for common continual learning benchmarks, the assumption of linear mode connectivity holds, and this section aims to push the linear mode connectivity to its boundaries in order to appropriately scope this interesting phenomenon that has significant consequences to understanding and development of continual and multitask learning.

## C  DETAILS OF EXPERIMENTAL SETUP

In this section, we answer two important questions regarding our experiments and result:

1. What decisions did we make for our experiments (e.g., continual learning setup, model architectures, benchmarks, etc.)?

2. Why did we make those decisions and how close our setup is to other established continual learning works?

To this end, we start with the first question by reviewing our experimental setup, followed by hyper-parameters we used in the main experiments. To answer the second question, in each sub-section, we show that our setup is close to several established works in the continual learning literature.

### C.1  BASELINES AND ARCHITECTURES

We chose the baselines and architectures based on three criteria.

The first criterion is the diversity of methods. In addition to the naive SGD, we have selected A-GEM (Chaudhry et al., 2018b) and ER-Reservoir (Chaudhry et al., 2019) as representative of episodic memory methods, EWC (Kirkpatrick et al., 2017) for regularization methods. The second criterion is reproducibility. While there are several other continual learning methods, we only reported the methods that we could replicate their results either by their official implementations or self-implementations. The final selection criterion is performance. A-GEM and ER-Reservoir are high-performance episodic memory methods. In addition, we have added the stable-SGD (Mirzadeh et al., 2020) as the current state of the art continual learning method, in addition to the previous state of the art methods such as A-GEM and ER-Reservoir.

At the end of this section, we show that by carefully choosing a similar continual learning setup to common setup in the literature, we have made it easier to compare our results with other works that we could not replicate ourselves.

### C.2  CONTINUAL LEARNING SETUP

In the MNIST and CIFAR experiment in Section 2, we have chosen 5 tasks so that we can show microscopic metrics (e.g., the evolution of accuracy for all tasks) thorough the continual learning experience. In this experiment, our goal was not to reduce catastrophic forgetting. In contrast, in that experiment, we have aggravated the catastrophic forgetting to show that CKA and Euclidean distance are not helpful, even if the amount of forgetting is high. More specifically, we have not tuned the hyper-parameters, and we increased the catastrophic forgetting by using a per task rotation of $22.5°$ in the rotation MNIST benchmark. Moreover, for MNSIT experiment, we have used a two-layer MLP with 100 ReLU units. The ResNet18 in that experiment is similar to the ResNet18 in Section 5. Finally, in this setup, we have used the learning rate of $0.1$ for MNIST dataset and $0.05$ for the CIFAR experiment with batch size of 64. Finally, in this experiment, the number of of epochs per task is 5. We discuss the rationale behind this setup in Appendix C.

In the comparison experiment in section 5.1, we have increased the number of tasks to 20 tasks, and the model is trained on each task for one epoch. This benchmark is more challenging since we used one example per class per task (i.e., 200 for rotated and permuted MNIST, and 100 for split CIFAR-100). Thus, the model is allowed to see the data only once, and hence, the setting becomes closer to the real-world online setting. For the CIFAR experiment, to make sure the first task solution is within a good basin, we train the only first task for five epochs in this experiment and continue the training for one epoch for the next tasks. While our results might seem initially lower than the results reported in other continual learning works, we emphasize that this is due to the limited episodic memory we have used in our setup. There are two reasons behind this choice of memory size and number of epochs. First, to make it closer to common setup in other related works such as A-GEM and ER-Reservoir that are state of the art in episodic memory methods. Second, the limited episodic memory and number of epochs allow measuring "how effective each continual learning method uses the previously seen data". In this experiment, we use a two-layer MLP with 256 ReLU units for MNIST datasets and ResNet18 with three times less feature for CIFAR dataset. We note that to have

a similar setup with our baselines, the task identifiers are used to select the correct output head only in CIFAR-100 experiments.

### C.3 HYPER-PARAMETERS

For the experiment in Section 5.1, we have used the following grid for each model. We note that for other algorithms (e.g., A-GEM, and EWC), we ensured that our grid contains the optimal values that the original papers reported.

#### NAIVE SGD

- learning rate: [0.25, 0.1, **0.01** (MNIST, CIFAR-100), 0.001]
- batch size: 10

#### EWC

- learning rate: [0.25, **0.1** (MNIST, CIFAR-100), 0.01, 0.001]
- batch size: [**10**, 64]
- $\lambda$ (regularization): [1, **10** (MNIST, CIFAR-100), 100]

#### A-GEM

- learning rate: [0.1, **0.1** (MNIST), **0.01** (CIFAR-100), 0.001]
- batch size: [**10**, 64]

#### ER-RESERVOIR

- learning rate: [0.25, **0.1** (MNIST), **0.01** (CIFAR-100), 0.001]
- batch size: [**10**, 64]

#### STABLE SGD

- initial learning rate: [0.25, **0.1** (MNIST, CIFAR-100), 0.01, 0.001]
- learning rate decay: [0.9, 0.85, **0.8**(CIFAR-100), **0.6**(MNIST)]
- batch size: [**10**, 64]
- dropout: [**0.25** (MNIST), **0.1** (CIFAR-100)]

#### MODE CONNECTIVITY SGD

To obtain continual minima (i.e., $\hat{w}_1$ to $\hat{w}_{20}$), we use the same hyper-parameters as the stable SGD. To minimize the objective function in Eq. (3) and obtain $\bar{w}_1$ to $\bar{w}_{20}$, we use the following grid:

- number of samples: [3, **5**, **10**, **20**]. We found that empirically more than five samples on each line would give sufficient information to minimize Eq. (3) for both MNIST and CIFAR experiments.
- learning rate: [0.001, **0.01** (CIFAR-100), **0.05**(MNIST), 0.1, 0.2 (MNIST, CIFAR-100)].

### C.4    How close is our experimental setup to the other works in the literature?

We end this section by showing that our setup is close to the common setup in the continual learning literature. More specifically, we discuss the following aspects of our setup:

**Benchmarks**: While we believe continual learning literature can benefit from a more diverse set of tasks such as reinforcement learning tasks, the majority of the focus in the field is devoted to computer vision tasks. One possible explanation is that despite having more control over the continual learning experience, and despite the fact that it is possible to train high-performance models in single task settings, still, models perform relatively poorly in continual learning setting. Hence, we followed the common benchmarks in the literature such as rotated and permuted MNIST, and split CIFAR-100. While we believe the community needs to agree on a more diverse set of benchmarks, the performance gap between continual learning minima and multitask minima is yet to be filled even on these simpler benchmarks.

**Architectures**: The choice of two-layer MLP with 256 hidden neurons for MNIST benchmarks and ResNet18 for CIFAR-100 benchmark is common in the literature (Chaudhry et al., 2018a; Mirzadeh et al., 2020; Chaudhry et al., 2019). We have used same architectures to make it easier to compare our results with other works as well.

### C.5    Implementation Details

In this section we discuss the visualization and implementation details.

#### C.5.1    How are surface planes generated?

To plot a surface plane (e.g., Fig. 3), we need three points to obtain two basis vectors. Suppose we have a set of three parameters $w_1$, $w_2$, and $w_3$ where each parameter is in vector in high-dimensional space. These vectors can be obtained by computing the flattened version of each layer in the neural network and then concatenating them. For the three mentioned points, we perform the following procedure:

1. Calculate two basis vectors: $\vec{u} = w_2 - w_1$, and $\vec{v} = w_3 - w_1$.

2. Orthogonalize the basis vectors by calculating $v = v - \cos(u, v)u$

3. Define a Cartesian coordinate system in the (x,y) plane that maps each coordinate to a parameter space by calculating $p(x, y) = w_1 + u \cdot x + v \cdot y$

4. Finally, for a defined grid on this coordinate system, we calculate the empirical loss of each $(x, y)$ coordinate using function $p(x, y)$.

Finally, we note that to generate plots that show loss along the interpolation between minima (e.g., Fig. 4), we simply calculate the interpolation solution between two minima and calculate the loss for that solution. For example, the minima on the interpolation line that connects $\hat{w}_1$ to $\hat{w}_2$ can be represented with $w_\alpha = \hat{w}_1 + \alpha(\hat{w}_2 - \hat{w}_1)$ where $0 \leq \alpha \leq 1$. We then calculate the empirical loss for $w_\alpha$.

### C.6    Reproducing MC-SGD

We provide an executable version of MC-SGD on the CodaLab platform. CodaLab keeps track of the full provenance generating a particular result, and thus, executable papers on this platform allow others to verify the results independently.

To this end, we provide five different runs of MC-SGD on rotation MNIST with 20 tasks to verify the results in Table 1. Each run uses a different random seed and can be fully replicated as it has all the required dependencies and the source code. Here are links to run 1, run 2, run 3, run 4, and run 5.

Each of the above links has a "stdout" section that logs the evolution of the average accuracy of MC-SGD. In addition, each run generates a "checkpoint" folder that includes weight files to MC-SGD weight files at the end of each task.

### C.6.1 MC-SGD Implementation

To implement MC-SGD, we use PyTorch (Paszke et al., 2019) because of its dynamic graph capability. The challenging part of the implementation is calculating Eq. (5), where we need to compute a summation of gradients on two lines. To this end, we first calculate the accumulated gradients over the mentioned line:

```python
def get_grads_on_interpolation_line(w_hat, w_bar, alphas, memory):
    """
    Get accumulated gradients on interpolation line
    between w_hat and w_bar
    """
    accumulated_gradients_on_line = 0.0

    # for each point on the interpolation line
    for alpha in alphas:
        interpolation_grads = []
        # (1) calculate the parameters on the interpolation line
        interpolation_w = w_hat + (w_bar - w_hat) * alpha
        model = make_model_from_param_vector(interpolation_w)
        loss = calculate_loss(model, memory)

        # (2) calculate the loss to obtain gradients
        loss.backward()
        # Now the gradients are stored in the graph

        # (3) collect the gradients
        for param in model.parameters():
            # get gradients for each module/block/layer
            interpolation_grads.append(param.grad.view(-1))
        interpolation_grads = torch.cat(interpolation_grads)

        # (4) accumulate gradients
        accumulated_gradients_on_line += interpolation_grads

    return accumulated_gradients_on_line
```

## D   Additional Results

### D.1   Conventional distance measures

In this section, we provide detailed $\ell_2$ distance and pair-wise CKA scores of the models introduced in Sec. 2.

As explained in Sec. 2, $\ell_2$ distance can be a misleading metric when it comes to comparing the multitask solution to the continual learning one. I.e., the continual solution on task 5 seems to be closer to the solution of task 1 than the multitask solution, even though the continual solution suffers from catastrophic forgetting. This could be indicative to the fact that compared to the pruning scenario, curvature might play a bigger role in continual learning and can not be ignored (e.g., the delta between parameters might align differently to the eigenvectors of the Hessian and of higher order derivatives).

However, the bound defined by Eq. (1) is not accurate enough if the vector $\hat{w}_2 - \hat{w}_1$ does not lie in the direction of the largest eigenvector. For instance, it might be the case (as we show later) that the optimization trajectory moves in a direction with low curvature (e.g., a low-loss valley), and thus the loss does not increase. The ineffectiveness of Euclidean distance is also verified empirically by middle column of Fig. 2. Even though the MTL minima for tasks are further from CL minima they have much higher accuracy. For the reference and further evidence, the full table of pairwise distances between all minima is shown in Fig. 12.

### D.2   All minima are linearly connected

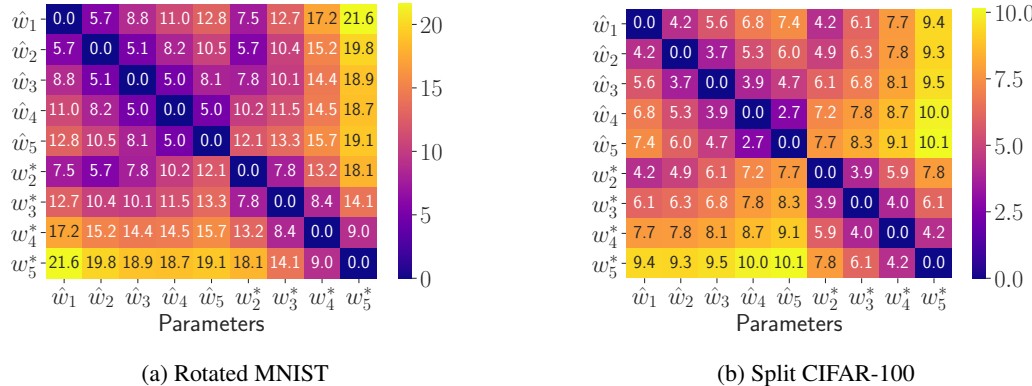

(a) Rotated MNIST

(b) Split CIFAR-100

Figure 12: Pairwise $\ell_2$ distances between all CL and MTL minima. For each CL minima, while MTL minima are further, they still are good minima due to the fact that they lie in a low-loss valleys, as we show in section 2.1.

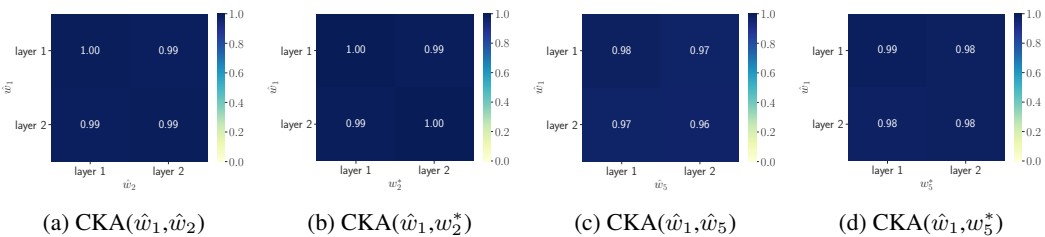

(a) CKA($\hat{w}_1,\hat{w}_2$)  (b) CKA($\hat{w}_1,w_2^*$)  (c) CKA($\hat{w}_1,\hat{w}_5$)  (d) CKA($\hat{w}_1,w_5^*$)

Figure 13: Rotated MNIST: Pairwise CKA similarity score between hidden layers

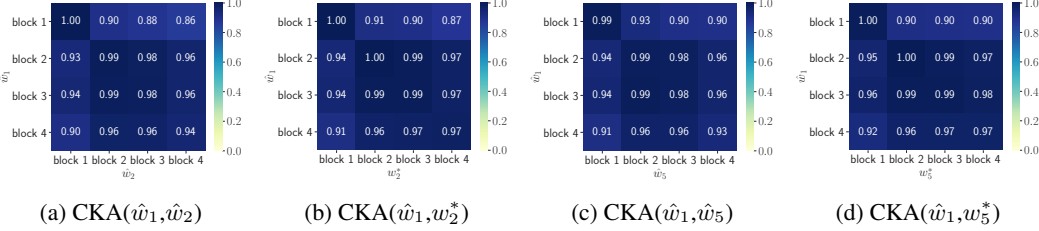

(a) CKA($\hat{w}_1,\hat{w}_2$)  (b) CKA($\hat{w}_1,w_2^*$)  (c) CKA($\hat{w}_1,\hat{w}_5$)  (d) CKA($\hat{w}_1,w_5^*$)

Figure 14: Split CIFAR-100: Pairwise CKA similarity score between ResNet blocks

In this section, we first show the extended learning setup for 3 tasks in Fig. 15. Eseentially, this is the extended version Fig.1 where an additional task is added.

Moreover, we provide a more detailed result for the interpolation plots in Fig. 4 where we show a zoomed version of those plots. For easier comparison, we repeat the plots from Fig. 4 here in Fig. 16b, 16d, 17b, and 17d with the same order. Fig .16a shows two paths (i.e., $\hat{w}_1$ to $\hat{w}_2$ and $\hat{w}_1$ to $w_2^*$) from Fig. 16b.

Although the path from $\hat{w}_1$ to $\hat{w}_2$ (in yellow) seems nearly non-increasing Fig. 16b, in Fig .16a we show that this path is in fact an increasing path while the loss on the interpolation path between $\hat{w}_1$ and $\hat{w}_2$ is

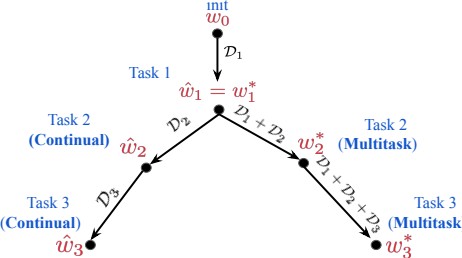

Figure 15: Setup for continual and multitask learning with 3 tasks

non increasing. Similarly, this phenomenon is shown in Fig. 16c for $\hat{w}_2$ to $\hat{w}_3$ and $w_3^*$ on rotated MNIST dataset and in Fig. 17 for split CIFAR-100 dataset.

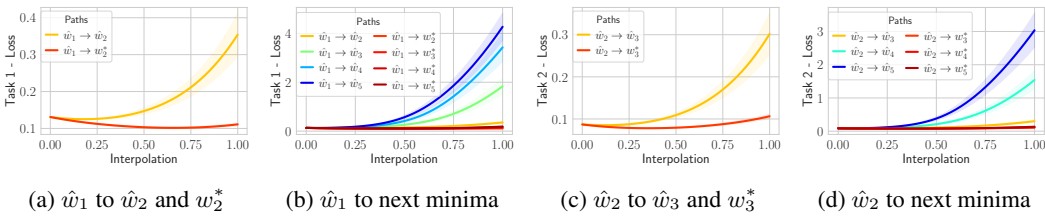

(a) $\hat{w}_1$ to $\hat{w}_2$ and $w_2^*$  (b) $\hat{w}_1$ to next minima  (c) $\hat{w}_2$ to $\hat{w}_3$ and $w_3^*$  (d) $\hat{w}_2$ to next minima

Figure 16: Detailed loss plots for interpolation paths between minima on rotated MNIST

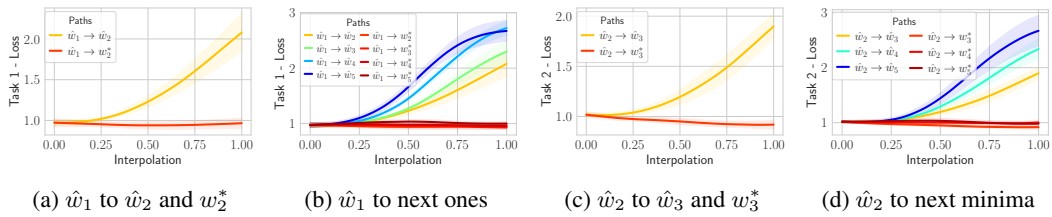

(a) $\hat{w}_1$ to $\hat{w}_2$ and $w_2^*$  (b) $\hat{w}_1$ to next ones  (c) $\hat{w}_2$ to $\hat{w}_3$ and $w_3^*$  (d) $\hat{w}_2$ to next minima

Figure 17: Detailed loss plots for interpolation paths between minima on split CIFAR-100

### D.3 STABLE SGD METHOD WITH MEMORY

One interesting question to pursue would be how would the stable SGD method perform if given the same size of episodic memory. To this end, we add an episodic memory to the stable-SGD method with similar size of MC-SGD (i.e., one example per class per task).

We show the performance of the stable SGD method in Table 2 on Rotated MNIST and Permuted MNIST with 20 tasks, with a similar setup in Sec. 5.1. We can see in Table 2 the additional episodic memory improves the average accuracy by 2.1% in Permuted MNIST and 2.5% in Rotated MNIST.

Table 2: Performance of the stable SGD method with access to episodic memory

|  | **Permuted MNIST** | | **Rotated MNIST** | |
|---|---|---|---|---|
| Method | Average Accuracy (%) | Forgetting | Average Accuracy (%) | Forgetting |
| Stable SGD | 80.1 ($\pm$ 0.51) | 0.09 ($\pm$ 0.01) | 70.8 ($\pm$ 0.78) | 0.10 ($\pm$ 0.02) |
| Stable SGD + episodic memory | 82.2 ($\pm$ 0.83 ) | 0.09 ($\pm$ 0.01) | 73.3 ($\pm$ 0.97) | 0.10 ($\pm$ 0.03) |

Moreover, in Fig. 18, we compare the loss on the interpolation paths between SGD, Stable-SGD with memory, and MC-SGD methods. To show Stable SGD minima, we use $\widetilde{w}$ notation. We can see that the additional episodic memory helped the stable-SGD method to gain a significant boost. Still, the regularization term in the MC-SGD method helps it find minima with better mode connectivity.

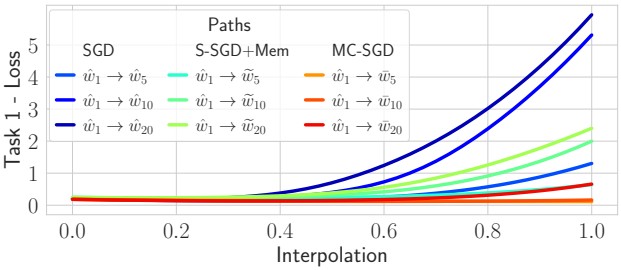

Figure 18: Interpolation plots for SGD, stable SGD with memory, and MC-SGD on Rotation MNIST benchmark with 20 tasks: Stable SGD with memory cannot find linearly connected minima.

### D.4 MODE CONNECTIVITY OF EWC MINIMA

One interesting question is that if regularization methods such as EWC can implicitly help obtain minima that are linearly connected. Similar to the Fig. 7 and Fig. 18, we show the interpolation paths for EWC minima In Fig. 19. We can see that although the EWC minima perform better than SGD, they are not linearly connected to the first task's minimizer.

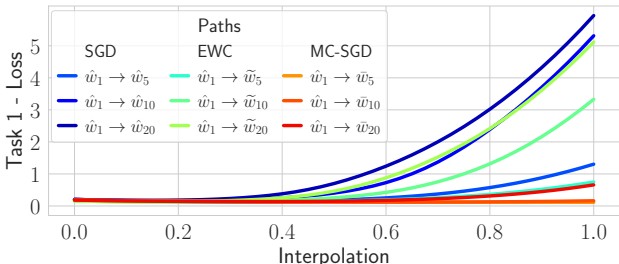

Figure 19: Interpolation plots for SGD, EWC, and MC-SGD on Rotation MNIST benchmark with 20 tasks: the EWC method cannot find linearly connected minima.

### D.5 PERMUTED MNIST WITH 50 TASKS

In this experiment, the MLP hidden layers increased from 256 to 512 for this experiment since the number of tasks increased. Moreover, the size of episodic memory is increased from 200 to 500 (one example per class per task).

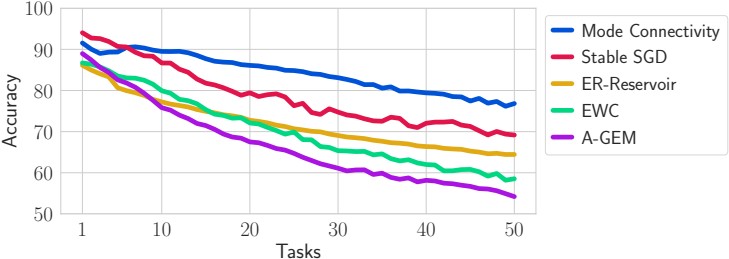

Figure 20: Evolution of the average accuracy for different methods on permuted MNIST with 50 tasks.

