# OpenReview forum: "Linear Mode Connectivity in Multitask and Continual Learning"
_ICLR.cc/2021/Conference — ICLR 2021 Poster_

### Official Review · AnonReviewer1 · 2020-10-27
**Connection between MTL and CL is very nice!**

**Rating:** 7
**Confidence:** 5

**Review:**

Summary

The paper studies the relation between the geometry of solutions of continual (CL) and multi-task learning (MTL). Towards this end, the authors empirically identify that all the solutions of CL (i.e. solutions obtained after each task) and MTL are connected by a linear region of low error. This is a very interesting finding and, to my knowledge, has not been studied previously in the CL literature. Based on this observation, the authors propose a memory and regularization-based CL method, MC-SGD, that ensures that the final CL solution is linearly connected to all the task’s solutions. The authors further demonstrate that the solution of the MTL lies in the region where the Hessian of the loss function is low and hence the regularization-based approaches that make use of curvature information (e.g.) EWC, are a promising direction for CL. Experiments are conducted on Permuted and Rotated MNIST, Split CIFAR benchmarks. MC-SGD performs strongly compared to other baselines.

Positives

1- I quite enjoyed reading the paper. It is very well-written and insightful.


2- Sections 2.1 and 3 are very nice. The finding, albeit empirical, that the solutions of multi-task and continual learning are linearly connected could prove to be very important for future research in CL.


3- Experimental results are very strong. I am frankly quite surprised that the gain on top of ER is that much. Although the authors mention it in Section 5 that they use a similar setup as in the other works, I just want to clarify the number of epochs here. If my understanding of their work is correct then Chaudhry et al., in all their work use a single-epoch setup where Farajtabar et al., used multiple epochs. Do you use single or multiple epochs?


Negatives

I don’t have any major concerns about the work except for a few nitpicks and questions.


1- See the multiple-epochs remark above.


2- Fig.5: Can you compute all the eigenvectors and show whether the cosine similarity in the eigenvectors corresponding to the smallest eigenvalues actually increase when you go towards the multi-task solution. You can reduce network size if compute is the problem.

3- Eq.5 (or similarly Eq. 3): It seems that one needs the solution of task t $\hat{w}_t$ for this loss to work. If one just receives the task t how would one obtain this solution? Do you do this in two steps? Where, in step 1, you just compute the $\hat{w_t}$ starting from $\bar{w}_{t-1}$, and then, in step 2, you use the $\hat{w_t}$ obtained from step 1 to compute the final $\bar{w}_{t}$?

4- Fig. 7: Might want to highlight in the legend which path is CL and which is MC.

5- Page 5, 6th to last line, ~oneself~ itself

---

> ### Author Response · Authors · 2020-11-21
> **We thank the reviewer for the helpful comments. We are glad that the reviewer enjoyed reading our work and found our results strong. Below, we aim to clarify some points further.**
>
> **(1) Do you use single or multiple epochs?**
>
> Generally, the number of epochs varies across experiments. For the experiments where we compare MC-SGD with other baselines (e.g., Section 5.1, and Appendix D.3, D.4, D.5), we used a single epoch setting to have a similar setup with Chaudhry et al.  In section 2, we have used five epochs to highlight the differences between MTL and CL minima. We have updated Appendix C.2 to prevent confusion.
>
> **(2) Can you compute all the eigenvectors and show whether the cosine similarity in the eigenvectors corresponding to the smallest eigenvalues actually increase when you go towards the multi-task solution.**
>
> We are not sure if we have completely understood the reviewer’s comment. Generally, the number of eigenvalues/eigenvectors of the Hessian is in the order of 10^6 even in the simple case of MNIST experiments. Even if we reduce the network size significantly, the number of eigenvalue/eigenvector still remains very large. Moreover, as shown in Fig. 5 (c), moving across low-curvature is not sufficient but a necessary condition for having a high-performance solution. In that figure, both MTL and CL minima do not lie in a subspace spanned by top eigenvectors, but we know that MTL and CL minima are very different in terms of accuracy. If this answer is not convincing we probably misunderstood the reviewers intention and we appreciate it if they further clarify this comment.
>
> **(3) Do you do this in two steps? Where, in step 1, you just compute the starting from , and then, in step 2, you use the obtained from step 1 to compute the final?**
>
> That's true. As the reviewer correctly mentioned, our algorithm operates in two steps. However, this does not mean that the algorithm's running time would be twice the SGD algorithm. The reason is that unlike the first step that needs the whole training data to compute w_t, the second step works with only a small episodic memory. For example, in the performed experiment in section 5.1, the size of episodic memory is one example per class per task (i.e., 200 for MNIST benchmarks and 100 for CIFAR-100 benchmark). As a result, the runtime will be very close to the SGD algorithm.
>
> We have updated the text with the minor edits suggested by the reviewer. Thanks for pointing out.

---

### Official Review · AnonReviewer2 · 2020-10-28
**The question of how MTL and CL solutions are connected is well motivated, answered satisfactorily, and leveraged to motivate an effective approach to CL**

**Rating:** 7
**Confidence:** 4

**Review:**

############## Summary ##############

This submission asks the question of how the minima found by batch multi-task learning compare to those of continual learning. It empirically finds the they are connected via linear interpolation through a manifold of low error, and leverages this fact to come up with a clever new algorithm for continual learning that performs better than various existing continual learning baselines on three benchmark data sets.

############## Strengths ##############

1. The question of how to connect multi-task to continual learning solutions is well motivated via simple introductory experiments.
2. The answer to this question, that there is a linear mode connectivity, motivates a simple, elegant, and effective algorithm.

############## Weaknesses ##############

1. The paper could benefit from substantial editing to make it clearer and easier to follow. I found myself having to re-read various sections to properly understand how the different parts of the paper were connected.
2. The empirical evaluation is done only on three benchmarks. It could be valuable to add evaluations on additional data sets, like Omniglot (https://github.com/brendenlake/omniglot).

############## Recommendation ##############

I recommend this paper for acceptance, but urge the authors to substantially revise their manuscript to make it more approachable. I believe this paper to be self-contained, with a clear question being asked, which hadn't been asked before: how are the solutions to multi-task and continual learning methods connected. The authors find that there is linear connectivity between these solutions, and use this fact to motivate a simple yet effective continual learning algorithm.

############## Arguments ##############

The question of whether and how the solutions to multi-task and continual learning are connected is highly relevant. While most prior literature had assumed that some distance metric in the parameter space was the correct way to measure their connection, this work is motivated by the experiments in Fig. 2, which show that these metrics are not quite appropriate. Instead, the authors show that linear mode connectivity better explains how multi-task and continual learning solutions are related.

The manuscript then deviates to an analysis of when this type of connectivity holds by analyzing second-order Taylor approximations. I had to re-read this section (Section 3) multiple times in order to find what the relevance of it was to the submission. My conclusion was that the point is that the fact that the parameter vectors move in directions of low curvature means that interpolation in those directions doesn't increase the loss by much. This fact seems to be somewhat hidden in the text. I encourage the authors to place emphasis on what their analysis is attempting to find before diving into it in depth, as it is easy to lose the reader if they are not aware of where the analysis is going from the start.

The proposed algorithm is clever and simple: it leverages past data not only to approximate the loss of the previous tasks on the new solution, but also to add a regularization encouraging a low-loss linear path between the solutions. Although the authors experiment with very few data sets, I believe they sufficiently show the applicability of their method and the fact that it performs well. It would be interesting to see how differently the method would perform if instead of the MC regularization, the authors used the EWC one. This would help avoid conflating the claim "regularization + replay is best" from "MC regularization + replay is best". Similarly, it would be relevant to reproduce Figure 7 with the solutions found by baselines, to assess whether they also find linear connectivity solutions. The claims would be stronger if the authors showed that baselines don't find linearly connected solutions.



############## Additional feedback ##############

The following points are provided as feedback to hopefully help better shape the submitted manuscript, but did not impact my recommendation in a major way.


Intro
- It seems like the authors interchangeably used en-dash and em-dash. They also used en-dash to open, but not close, a statement.
- What confounding factors are removed by doing w1 --> w1,w2 other than initialization? The text makes it sound like there's more but no other is discussed.
- Contribution 3: benchmark --> benchmarks
- I believe compressed related work sections or those pushed to the appendix make it hard to place the contribution in context. I encourage the authors to expand this section in the main paper.

Sec 3
- I had two main questions when reading this section:
    - Why doesn't EWC find such a low curvature path, if it precisely penalizes deviations in directions of high curvature?
    - Why can't we just use the proper Taylor expansion instead of Euclidean distance then, to measure forgetting, instead of mode connectivity?
    -These two questions were answered towards the end of the section by showing that this is not a sufficient condition, and are then explicitly addressed by suggesting that second-order approximations are a promising direction for future work. I encourage the authors to clarify this before diving into the analysis, so the reader knows what to look for when reading this section.
- The caption for Fig. 5 doesn't explain difference between b and c, which is only somewhat explained in text later.

Sec 4
- Regularization only considers low-loss path between the solution to the immediately previous task and the current solution (but not the solutions to all past tasks), assuming that the immediately previous solution contains sufficient information. Was this empirically tested? The EWC authors claim that using only the previous model in their setting is insufficient [1], so it would be interesting to see if there's a similar effect here.

Appendices
- Very complete: additional results, justification of experimental setting.

Style, grammar:
- appendix X --> Appendix X
- second order Taylor --> second-order Taylor expansion/approximation
- minima is often used as a singular, which should be minimum
- regularization based --> regularization-based
- rehearsal based --> rehearsal-based
- Inconsistent italization of i.e.
- few shot learning --> few-shot learning


[1] Kirkpatrick, J., Pascanu, R., Rabinowitz, N., Veness, J., Desjardins, G., Rusu, A. A., ... & Hassabis, D. (2018). Reply to Huszár: The elastic weight consolidation penalty is empirically valid. Proceedings of the National Academy of Sciences, 115(11), E2498-E2498.
Chicago

---

> ### Author Response · Authors · 2020-11-21
> **We thank the reviewer for valuable comments and helpful feedback. We are encouraged that the reviewer finds our method elegant and effective. Here, we clarify some points the reviewer mentioned.**
>
> **(1) It would be relevant to reproduce Figure 7 with the solutions found by baselines, to assess whether they also find linear connectivity solutions**
>
> We have added two new experiments (Appendix D.3 and D.4). In Figure 18, we have shown the loss on the interpolation paths between stable SGD minima, and in Figure 19, we have shown the same graph for the EWC method. Thank you for your comment. As the reviewer mentioned, these additional results would make our claim stronger.
>
> **(2) Intro: What confounding factors are removed other than initialization?**
>
> As noted by the reviewer the direct confounding factor is the initialization. But we used this general term since it implicitly includes the initialization and its impact on the minima (e.g., optimization trajectory).
>
> **(3) Intro: dashes and typos**
>
> We have updated the text with the edits reviewer kindly suggested.
>
> **(4) Intro: compressed related work sections**
>
> Thanks for the suggestion. We have moved the entire related work to the main body.
>
> **(5) Sec 3: I encourage the authors to clarify this before diving into the analysis, so the reader knows what to look for when reading this section.**
>
> Thanks for this comment. It was indeed a very helpful feedback. We have significantly revised section 3 and further clarify the intention. As correctly realized by the reviewer, our intention was to highlight why existing methods such as EWC that are also trying to penalize the change in directions of high curvature fail to find a plausible direction. The short answer is that they only rely on the second-order Taylor approximation of the loss function, which may be easily violated. More details are provided in the revised version of section 3.
>
>
> **(6) Sec 3: The caption for Fig. 5**
>
> Updated. Thanks for pointing out.
>
> **(7) Sec 4: Regularization only considers low-loss path between the solution to the immediately previous task and the current solution (but not the solutions to all past tasks), assuming that the immediately previous solution contains sufficient information. Was this empirically tested?**
>
> This is a very interesting question! In fact, we empirically tested this in our initial steps during feasibility study. More especially, on the rotated MNIST benchmark with 5 tasks we found that the gain is less significant and is not worth the additional complexity in implementation and memory requirement. The proposed MC-SGD method needs only two solutions (w_{t}, and w_{t-1}), while in the latter case, the method needs all the previous minima, and the memory requirement would grow linearly with the number of tasks, which is a significant drawback. That’s why we have not pursued this direction in our scaled experiments.
>
> **(8) Style, Grammar**
>
> Thank you for the suggestions. We have updated the text.
>
> **(9) Presentation of Section 3**
>
> Thank you for your feedback. We have revised section 3 to prevent the confusion.
>
> **(10) The empirical evaluation is done only on three benchmarks. It could be valuable to add evaluations on additional data sets, like Omniglot.**
>
> We agree with the reviewer on this comment. However, we would like to explain why we did not have reported the results on Omniglot:
> Unfortunately, the Omniglot benchmark is not commonly used in the literature, compared to MNIST and CIFAR-100 benchmarks. In our case, none of our baselines have used this benchmark on their papers.
> To the best of our knowledge, the literature has not agreed on a specific setup on Omniglot. The setups are not consistent between those few works that report results on this benchmark. For instance, looking at the papers in ICLR2020, Adel et al. [1] used Omniglot with 50 tasks with a 4-layer CNN architecture and there is not open access code and implementation to infer further details.  In contrast, Yoon et al. [2] use a modified Omniglot with 100 tasks using a modified version of LeNet. The code for this paper is not published to find further details.
> These challenges made us report the result on Permuted MNIST with 50 tasks instead. However, we appreciate this comment and we do our best to report result on an additional benchmark in future versions. But we believe this would take some time, probably going beyond the discussion period.
>
> [1] Adel, Tameem, et al. “Continual Learning with Adaptive Weights (CLAW).” ICLR 2020 : Eighth International Conference on Learning Representations, 2020.
>
> [2] Yoon, Jaehong, et al. “Scalable and Order-Robust Continual Learning with Additive Parameter Decomposition.” ICLR 2020 : Eighth International Conference on Learning Representations, 2020.

---

> > ### Comment · AnonReviewer2 · 2020-11-23
> > **Thanks for your response. I maintain that this is a strong contribution.**
> >
> > The authors provided an adequate response to most of my points. In particular, I am glad they included Figures 18 and 19 in the Appendix, demonstrating that EWC and stable SGD do not find linearly connected solutions, while their proposed MC-SGD does. The writing of Section 3 also seems to have been much improved.
> >
> > However, my concern about the number of benchmarks remains. I don't believe that the lack of consistency of existing work benchmarking on Omniglot (or any other data set) prevents an accurate comparison, especially with baselines like A-GEM and ER-reservoir, for which there is open-sourced code available.
> >
> > In spite of this, the paper remains a solid contribution, with very good motivation and a strong proposed algorithm.

---

### Official Review · AnonReviewer4 · 2020-10-28
**The paper studies continual learning from the perspective of multi-task learning and shows that a linear path of low error regime connects the found local minima of the subsequent tasks  with that of multi-task learning.**

**Rating:** 7
**Confidence:** 5

**Review:**

The paper starts by that observing the local minima obtained in a multi task scenario are connected with a linear path of low error regime to the local minima of each task in a continual learning scenario in contrast to the path between the different minima of tasks incrementally learned, provided the both training of multi task and continual learning have started from the same initialization. The paper studies and shows this mode connectivity empirically. It further discusses and analyses the factors behind this connectivity while noting that this is valid when tasks have shared structure in which local minima can be found nearby.
Motivated by these observations, the paper proposes a new solution to the continual learning problem. This is done by defining a new loss that forces this connectivity between the minima of  the previous task and the current task. As this requires evaluating the loss of a previous task, an experience replay of stored previous samples is used.  The paper shows improved performance in comparison to existing methods on different benchmarks of 20 tasks long each.
While I enjoy reading the paper, I think the clarity of the text can be enhanced specifically when referring to figures. The second reference to figure 2, comments on the Euclidean distance without explaining where this is shown in the figure and that was not so clear in the figure caption either. Figure 7, it is not clear what corresponds to the Naïve SGD and what corresponds to the MC SGD.
On the empirical evaluation, I wonder how stable SGD would perform if was given access to the same replay buffer?  It would be also interesting to show the comparison of the path with Stable SGD since it is supposed to find wider local minima where other tasks minima are likely to be nearby.
I assume that split cifar 100 is multi-head, would the proposed solution shows similar advantages in the shared head scenario?

---

> ### Author Response · Authors · 2020-11-21
> **We thank the reviewer for the insightful comments. We are pleased that R4 enjoyed reading our work. Below, we provide some clarification on reviewer comments and questions.**
>
> **(1) The clarity of the text can be enhanced specifically when referring to figures**
>
> We have updated the text and have added further explanation regarding the figures (e.g., Figure 2 and Figure 7) and their captions. Thank you for your comment.
>
> **(2) How stable SGD would perform if was given access to the same replay buffer? It would be also interesting to show the comparison of the path with Stable SGD**
>
> We have included a new section (Appendix D.3) where we have compared the performance of Stable SGD with and without replay buffer.
> Moreover, we have included a new figure in the mentioned appendix to compare the loss of interpolation paths for SGD, Stable SGD, and MC-SGD. In the figure, we can see that by adding the episodic memory, Stable SGD improves, but MC-SGD minima are still better connected, perhaps due to leverating mode connectivity prior.
>
> **(3) I assume that split cifar 100 is multi-head**
>
> True. Our setup follows the setup of our baselines (e.g., A-GEM, ER), where, the task identifiers are used to select the correct output head in the CIFAR experiment.  This is explained in Appendix C.
>
> **(4) Would the proposed solution shows similar advantages in the shared head scenario**
>
> While it is interesting to perform an experiment with a shared single head setup (incremental domain learning) in the future, the most popular setting in CIFAR100 benchmark uses separate heads (incremental task learning). This  includes all the baselines that we have compared against them too. Moreover, it’s noteworthy that the MC-SGD method works directly with the parameters of the networks (or a subnetwork like the shared bottom) and can be independent of the heads. We have updated the Appendix C for further clarification.

---

### Author Response · Authors · 2020-11-21
**We thank all the reviewers for their time and effort, resulting in insightful reviews of our work with helpful comments and feedback. We are pleased that all reviewers recommend our paper for acceptance. We are also encouraged that they enjoyed reading our work (R1, R4), found our algorithm simple, elegant, and effective (R2), with strong experimental results (R1).**

**Summary of the Updates**

In this revision, we have made the following updates to improve the quality of our submission further:

**(1)** We have updated the text with the feedback we received from the reviewers.  More specifically, we have updated the explanation of figures (R2 and R4),  moved the entire related work back to the main paper (R2), significantly revised and improved Section 3 (R2), and incorporated minor edits suggested by all reviewers.

**(2)** We have added two new experiments in the appendix.

- In the first experiment (Appendix D.3), we have added the results for the stable SGD method with the access to replay buffer (R4) and the interpolation plots for stable SGD with memory (R4, R2). We show that although the performance of stable SGD improves, it is still suboptimal compared to the MC-SGD algorithm. Interestingly, the minima found by stable SGD are not linearly connected and thus it is not able to leverage the mode connectivity to mimic the multitask solution.

- In the second experiment (Appendix D.4), we have added the interpolation plots for the EWC to show that the minima found by EWC regularization are not linearly connected. Moreover, we highlighted at the beginning of section 3 why existing methods such as EWC that are also trying to penalize the change in directions of high curvature fail to find a plausible direction. The short answer is that they only rely on the second-order Taylor approximation of the loss function, which may be easily violated. Note also EWC uses the diagonal of the Hessian to approximate the true curvature which adds another source of error. More details are in the revised version of section 3.

---

### Decision · Program_Chairs · 2021-01-07
**Final Decision**

**Decision:**

Accept (Poster)

**Comment:**

The paper is presenting an important empirical finding. When the learning algorithms are initialized with the same point, the continual and multitask solutions are connected by linear and low-error paths. Motivated by this finding, the paper proposes a new continual learning algorithm based on path regularization. The paper received unanimously good scores. I agree with the reviews and recommend acceptance.